# NEMt: Fast Targeted Explanations for Medical Image Models via Neural Explanation Masks

Bjørn Møller[*†1], Sepideh Amiri[†1], Christian Igel[1], Kristoffer Knutsen Wickstrøm[2], Robert Jenssen[1,2,3], Matthias Keicher[4], Mohammad Farid Azampour[4], Nassir Navab[4], and Bulat Ibragimov[1]

[1]Department of Computer Science, University of Copenhagen, Copenhagen, Denmark
[2]Department of Physics and Technology, UiT The Arctic University of Norway, Tromsø
[3]Norwegian Computing Center, Oslo, Norway
[4]Technical University of Munich, Munich, Germany
[†] indicate equal contribution

## Abstract

A fundamental barrier to the adoption of AI systems in clinical practice is the insufficient transparency of AI decision-making. The field of Explainable Artificial Intelligence (XAI) seeks to provide human-interpretable explanations for a given AI model. The recently proposed Neural Explanation Mask (NEM) framework is the first XAI method to explain learned representations with high accuracy at real-time speed. NEM transforms a given differentiable model into a self-explaining system by augmenting it with a neural network-based explanation module. This module is trained in an unsupervised manner to output occlusion-based explanations for the original model. However, the current framework does not consider labels associated with the inputs. This makes it unsuitable for many important tasks in the medical domain that require explanations specific to particular output dimensions, such as pathology discovery, disease severity regression, and multi-label data classification. In this work, we address this issue by introducing a loss function for training explanation modules incorporating labels. It steers explanations toward target labels alongside an integrated smoothing operator, which reduces artifacts in the explanation masks. We validate the resulting Neural Explanation Masks with target labels (NEMt) framework on public databases of lung radiographs and skin images. The obtained results are superior to the state-of-the-art XAI methods in terms of explanation relevancy mass, complexity, and sparseness. Moreover, the explanation generation is several hundred times faster, allowing for real-time clinical applications. The code is publicly available at //github.com/baerminator/NEM_T.

## 1 Introduction

Explainable Artificial Intelligence (XAI) in the medical area is motivated by the need for trust and transparency in AI-driven medical diagnosis, prognosis, treatment planning, and report generation [1, 2]. This need is steadily increasing due to the flourishing development of deep learning (DL) models in these areas [3]. In contrast, existing XAI methods suffer from limitations that impair their usefulness in clinical medical image analysis. Many methods focus too much on individual input features (e.g., pixels) and fail to detect complex patterns, while mask-based techniques optimized for finding regions in the input image are very slow [4, 5]. The recently proposed **N**eural **E**xplanation **M**asks (NEM) framework [5] has been shown to produce accurate mask-based explanations when applied to the representations of models trained in an unsupervised manner with a latency allowing for real-time inference. Currently, it does not take image labels into account. Therefore, it cannot explain a particular output component, for instance, answer the question of which parts of the input affect the score for a particular class (e.g., a pathology). This limits its application in the medical domain. In this study we

- introduce **N**eural **E**xplanation **M**asks with **t**arget labels (NEMt), a new NEM loss function for learning explanations of classifiers in the supervised setting;

- propose *stochastic weighted neighborhood averaging*, a convolutional layer in which parts are randomly masked out, to remove artifacts in DL-based occlusion mask prediction;

- evaluate NEMt and compare it to state-of-the-art XAI methods in the context of explaining classifiers for dermatoscopic skin and X-ray lung data, demonstrating that NEMt efficiently generates accurate, compact explanations;

- adapt the NEM framework [5] for unsupervised medical data analysis, and evaluate it for explaining representations learned by MedCLIP [6].

*Corresponding Author.

Proceedings of the 6th Northern Lights Deep Learning Conference (NLDL), PMLR 265, 2025.

## 2 Related work.

Feature attribution and set-wise explanation are two distinct directions in XAI that interpret the explanation task differently. Feature attributions estimate the individual contribution of each input feature to the model prediction for a particular sample [7–12]. Examples are gradient-based sensitivity analyses such as Grad-CAM (Gradient-weighted Class Activation Mapping [10]), methods based on SHAP (Shapley Additive exPlaination [9]), and LIME (Local Interpretable Model-agnostic Explanations [8]). However, explanations relying on feature-based attribution may be misleading when features have strong correlations or complicated interactions that are difficult for individual features to capture. In contrast, set-wise approaches aim at identifying subsets of informative features that have a collective impact on the model's prediction [13, 14]. This is especially helpful for comprehending intricate decision-making processes where the combination of features, rather than individual attributes, drives the prediction. However, these methods often suffer from long processing times and potentially produce poor-quality explanations [15].

Most XAI methods are designed for the supervised setting in the sense that they generate explanations for model outputs such as classifications, where the explanation is provided for an input-output pair similar to those used for supervised training of the model. However, recent advances in unsupervised deep learning [16, 17] have increased the need for XAI methods working for the unsupervised setting leading to many new contributions in this field [12, 18, 19]. These methods seek to explain what part of an input predominantly determines its (latent) representation learned by a (DL) model. Here the most prominent approach is RELAX and its variants [12], which provides feature-attribution explanations and has successfully been applied to medical imaging [20]. Recently, the Neural Explanation Mask (NEM) framework [5] was proposed for explaining unsupervised XAI and it was shown to compare favourably to previous unsupervised XAI methods, such as RELAX, in terms of the generated explanation masks while being significantly faster. The NEM framework aims to distinguish between significant and insignificant input regions. This approach is similar to methods from the field of Weakly Supervised Semantic Segmentation (WSSS), which extract segmentation labels from learned model representations, as exemplified by the works of Kweon et al. [21] and Araslanov and Roth [22].

## 3 Methodology

In this section, we start by presenting the background on the NEM framework, and then introduce our proposed NEMt. Finally, we outline our new approach for artefact removal through local stochastic smoothing.

### 3.1 NEM

The NEM framework [5] transforms a given differentiable ML model $\Phi$, referred to as the *frozen network*, into a self-explaining system by augmenting it with a neural network-based explanation module $\Psi$. The module is trained to generate occlusion-based explanations for $\Phi$. More specifically, for a given input $x$, an explanation mask $m = \Psi(x)$ is generated concurrently with the model output $y = \Phi(x)$. The mask $m$ serves to pinpoint the parts of $x$ that influence $y$ most. $\Psi$ is trained on (unlabeled) training images $\mathbf{X}$ to directly predict $m$ instead of performing some post-training optimization over $m$, which was the path in the existing occlusion-based explanation methods [13, 14, 23]. Figure 1 illustrates the overall framework and training process.

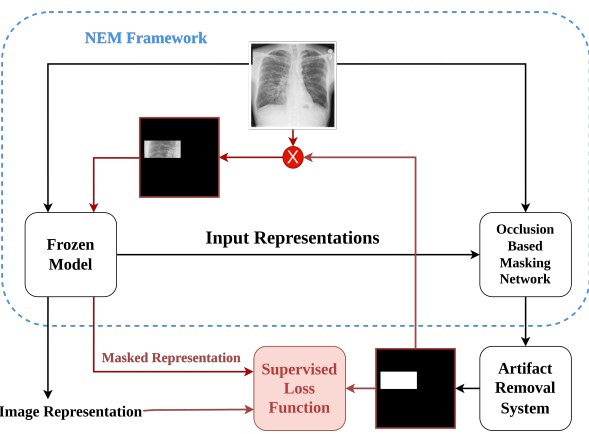

**Figure 1.** Overview of NEMt. The paths highlighted in black are used during the training and inference phase, and red paths indicate the flow that occurs exclusively during the training phase. The blue section shows the NEM framework.

The choice of the neural architecture of $\Psi$ and how it is attached to the frozen model $\Phi$ leads to different instantiations of the NEM framework. One specific instantiation is NEM-U. By considering mask-based explanation as segmenting images into areas of importance and non-importance, NEM-U draws on U-Net [24], which is commonly used to handle segmentation tasks. More specifically, the NEM-U treats the frozen model $\Phi$ and the masking model $\Psi$ as the encoder and the decoder of U-Net, respectively, and leverages on skip connections to extract intermediate representations from $\Phi$.

To train a NEM model, we need a loss function that defines what should be highlighted in a given input. In the case of unsupervised representation learning, a natural goal is to segment an image $x$ into sections that either do or do not strongly affect

the learned latent representation $\Phi(x)$. This can be recast as finding the sparsest mask, such that the model representation of the masked image is still close to the model representation of the original. This leads to a loss of the form

$$\mathcal{L}(\mathbf{X}) = \sum_{x \in \mathbf{X}} d\Big(\Phi(x), \Phi\big(\Psi(x) \odot x\big)\Big)$$
$$+ \lambda_1 \big\|\Psi(x)\big\|_1 + \lambda_2 B(x, \Phi, \Psi), \quad (1)$$

where the $d$ is some distance function, $B(x, \Phi, \Psi)$ is some regularization term enforcing binary masks, $\|\cdot\|_1$ is the $l_1$ norm, $\odot$ is the hadamard product, and $\lambda_1, \lambda_2$ are hyper-parameters. We optimize over a range of inputs $\mathbf{X}$ to allow $\Psi$ to generalize to unseen data. We adopt the heuristic described in [5] and set

$$B(x, \Phi, \Psi) = -d\Big(\Phi\big(\Psi(x) \odot x\big), \Phi\big((1-\Psi(x)) \odot x\big)\Big)$$
$$+ d\Big(\Phi(x), \Phi\big(\Psi(x) \odot x\big)\Big).$$

## 3.2 NEMt

The NEM framework outlined in Section 3.1 was proposed for explanations in unsupervised scenarios. Here, we extend it to the supervised setting. We assume classification into $C$ classes $\{0, C-1\}$. Let $c$ be the class label predicted for an image $x$. We assume that the frozen model outputs logits $\Phi(x) \in \mathbb{R}^C$ and predicts the label $c = \arg\max_i \Phi(x)_i$, where $\Phi(x)_i$ is the $i$'th component of $\Phi(x)$ corresponding to the logit for class $i$. In the unsupervised setting, we determine which parts of $x$ strongly affect the whole learned representation $\Phi(x)$, while in the supervised setting we are only interested in what explains $\Phi(x)_c$.

### 3.2.1 Loss function for explaining specific output properties.

In the classification setting, we only consider the effects on the scalar $\Phi(x)_c$ measured by the squared $l_2$ norm. The loss function will be therefore

$$\mathcal{L}(\mathbf{X}) = -\sum_{\substack{x \in \mathbf{X} \\ c = \arg\max_i \Phi(x)_i}} \big(\Phi(x)_c - \Phi\big(x \odot \Psi(x)\big)_c\big)^2 + \lambda_1 \|\Psi(x)\|_1.$$
$$(2)$$

The loss is independent of the output dimensionality as it only considers the target logit. Compared to (1), the loss function is a negative sum, aiming to identify the smallest area that determines the label. In other words, we wish to find the smallest input perturbation yielding the largest output change. We have removed the regularization term $B(x)$, since we found a more robust method of deleting artifacts in the supervised case as described in the following section. This loss formulation allows for explaining a specific output channel $t$ by setting $c = t$ instead of $c = \arg\max_i \Phi(x)_i$.

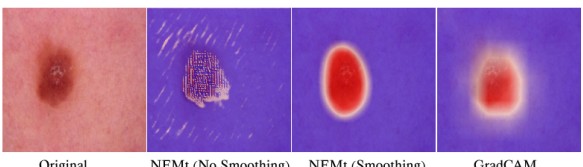

Original    NEMt (No Smoothing)    NEMt (Smoothing)    GradCAM

**Figure 2.** Removing artifacts via a stochastic weighted neighborhood averaging operator. The second image displays the result of the NEMt model without the smoothing filter. The third image shows the result when the smoothing filter is integrated into the explanation module $\Phi$. The fourth image shows a GradCam [25] explanation for comparison.

### 3.2.2 Explanation artifact removal via stochastic weighted neighborhood averaging.

Initial experiments with (2) showed that the method is more prone to generate artifacts in its explanation mask compared to what we observed in the unsupervised setting. See Figure 2 for an example. This is likely due to the fact that the term $d\big(\Phi(x), \Phi\big(x \odot \Psi(x)\big)\big)$ in the unsupervised loss function, which enforces closeness in representation space, also has a regularizing effect. In contrast, the supervised loss function rewards larger changes, potentially also in latent intermediate representations. It is not straightforward to employ a similar regularization term in the supervised setting, and our initial attempts to do so did not lead to satisfactory results. Thus, we opted to regularize in a different manner.

Our idea to address this issue was to bias the model to find explanations where the neighborhood of included pixels is also included in the explanation, that is, isolated pixels and other artifacts in the mask should be removed. To this end, we add a stochastic smoothing operator to the output of the explanation module $\Psi$. For each pixel in the network explanation mask, the filter performs a weighted average of the pixel and the elements of its local neighborhood via a convolution operation. To ensure faithful explanations, we train the network with the smoothing filter applied during training, rather than as a post-processing step. To increase the chance for all members of the neighborhood of a given important pixel to be included, in each forward pass random parts of the filter are "smoothly" masked out. This is done by randomly picking weights for each filter component so they are positive and sum to 1. A visualization of the effect of the filter can be seen in Figure 2.

## 4 Experimental Setup

This section outline the evaluation of our proposed methodology. First, we introduce the datasets, then

key implementation details and finally metrics used for comparisons to other methodologies.

## 4.1 Datasets

We used the 2018 RSNA Pneumonia Detection Challenge dataset that consists of 30,000 frontal chest radiographs from 6012 subjects with normal lungs, pneumonia-infected lungs, and other non-pneumonia abnormalities [26]. Each pneumonia pocket was annotated with a manually-drawn bounding box. We also used the HAM10000 dataset, comprising 10,015 dermatoscopic images divided into seven diagnostic categories and provides binary masks [27]. For evaluation, we extracted 1000 images from each dataset. For training the NEMt explanation modules, we extracted 21347 images from the RSNA dataset and 8012 images from the HAM10000 dataset, ensuring no overlap between patients in the training and test sets.

## 4.2 Implementation Details

To explore the feasibility of NEM-U for medical image analysis, we leveraged a pre-trained variant of the MedCLIP model [6] using a RESNET50 [28] architecture. We applied the MedCLIP model to the RSNA pneumonia dataset and explained the generated latent representations. We compared our method to RELAX [12] and its variant, U-RELAX, where RELAX explanations were thresholded based on uncertainty estimates.

To judge the performance in the supervised setting, we explained two classifiers based on the RESNET50 Architecture. The models are trained on the RSNA dataset and the HAM10000 dataset, respectively, using the same training splits as used for training the NEMt models. Given our goal is to identify areas of illness, we explain the channel belonging to the true target label using the targeted variant of (2). The RESNET50 architecture is extracted from the Pytorch Image Models library [29]. We compare our methodology against Integrated Gradients [30], Gradient SHAP [9], GradCAM++ [25], RISE++ [31], and Smooth Pixel Mask [23] with a area constraint of 5%.

We implemented all NEMt and NEM-U explanation architectures using U-Net decoders with 9 million trainable parameters. We applied the stochastic smoothing filter described in Section 3.2 with a neighborhood size of $(21, 21)$ when running NEMt in the supervised case. For the loss function for explaining representations, we chose the distance function $d$ to be negative cosine similarity (see [5]). In (1) we set $\lambda_1 = \frac{1}{8}$ and $\lambda_2 = \frac{1}{4}$ as is done in prior work [5]. In (2), we set $\lambda_1 = \frac{1}{2}$ based on visual inspection of the masks generated on the validation split of the training data. All models were trained with the Prodigy optimizer [32]. We provide an ablation study of various implementation choices in appendix A.3.

## 4.3 Metrics

For evaluation, we computed the complexity [33] and sparseness [34] metrics. These metrics favor explanations that highlight the minimum number of important pixels. Furthermore, we calculated *relevance rank* and *relevance mass* [35]. Relevance mass is determined by calculating the ratio of positive attributions located inside the ground truth bounding box to the total positive attributions. A high relevance mass score shows that the explanation significantly focuses on areas highlighted by human annotations while minimizing the focus on irrelevant areas. Relevancy rank is measured as the Intersection over Union (IoU) of the $k$ highest rated pixels according to attribution and the ground truth bounding box, where $k$ is equal to the size of the bounding box. Thus, the relevancy rank measures the overall ranking ability of a given explanation method. The metrics implementation was taken from the Quantus library [36]. We also considered the faithfulness [37] metric, which is only designed for labeled data. The idea of faithfulness is to assess whether pixels that are considered important are indeed important. This is achieved by iteratively removing input features and examining the model prediction and correlating the attributions' absolute values with the uncertainty in probability estimates. Finally, we evaluated the computation time per image by generating 1000 explanations and measuring the mean generation time. All attribution maps was min-max normalized before calculating experimental results to ensure consistency across methods.

## 5 Results

Table 1 and Table 2 summarize the results for explaining representations and classifications, respectively, showing locality metrics (relevance rate and relevance mass) and complexity metrics (complexity and sparseness) along with runtime performance. Additionally, in the supervised setting, the explanations are assessed by the faithfulness metric. When explaining representations, our method outperformed the other two approaches in all metrics except relevance rank: NEMt achieved the best performance in terms of relevance mass and runtime in both tasks and in terms of relevance rank in RSNA. It also achieved the second best results in terms of relevance rank in HAM10000 and for all tasks in terms of complexity and sparseness. Looking at faithfulness, NEMt performs best for pneumonia and third for skin cancer. Examples of explanations generated by the different approaches are given in Figure 3 and Figure 4.

**Table 1.** Comparison of XAI methods in the unsupervised setting using the RSNA dataset and a pre-trained Med-CLIP model.

| Method | Relevance Rank ↑ | Relevancy Mass ↑ | Complexity ↓ | Sparseness ↑ | Time (s) ↓ |
|---|---|---|---|---|---|
| RELAX [12] | **0.290** | 0.115 | 10.657 | 0.307 | 1.516 |
| U_RELAX [12] | 0.171 | 0.123 | 9.918 | 0.592 | 1.516 |
| **NEMt** | 0.204 | **0.212** | **8.295** | **0.937** | **0.003** |

**Table 2.** Comparison of XAI methods in the supervised setting. Results for the RSNA dataset in the upper table whereas the results for the HAM10000 is seen in the lower table. All experiments using a RESNET50 model trained on the dataset.

| Method | Relevance Rank ↑ | Relevancy Mass ↑ | Complexity ↓ | Sparseness ↑ | Faithfulness ↑ | Time (s) ↓ |
|---|---|---|---|---|---|---|
| Smooth Pixel Mask [23] | 0.098 | 0.095 | **8.977** | **0.868** | 0.024 | 3.663 |
| GradCAM [25] | 0.202 | 0.141 | 10.257 | 0.561 | 0.377 | 0.055 |
| Grad Shape [9] | 0.073 | 0.070 | 10.156 | 0.592 | 0.026 | 0.006 |
| Integrated Gradients [30] | 0.080 | 0.075 | 10.165 | 0.590 | 0.016 | 0.049 |
| RISE [31] | 0.124 | 0.077 | 10.686 | 0.274 | 0.240 | 7.368 |
| **NEMt** | **0.215** | **0.192** | 9.343 | 0.744 | **0.419** | **0.003** |

| Method | Relevance Rank ↑ | Relevancy Mass ↑ | Complexity ↓ | Sparseness ↑ | Faithfulness ↑ | Time (s) ↓ |
|---|---|---|---|---|---|---|
| Smooth Pixel Mask [23] | 0.495 | 0.582 | **8.690** | **0.907** | 0.399 | 3.663 |
| GradCAM [25] | **0.757** | 0.615 | 10.024 | 0.649 | **0.443** | 0.055 |
| Grad Shape [9] | 0.438 | 0.381 | 10.310 | 0.529 | -0.011 | 0.006 |
| Integrated Gradients [30] | 0.461 | 0.395 | 10.298 | 0.533 | 0.013 | 0.049 |
| RISE [31] | 0.271 | 0.279 | 10.675 | 0.290 | -0.009 | 7.368 |
| **NEMt** | 0.723 | **0.838** | 8.855 | 0.828 | 0.298 | **0.003** |

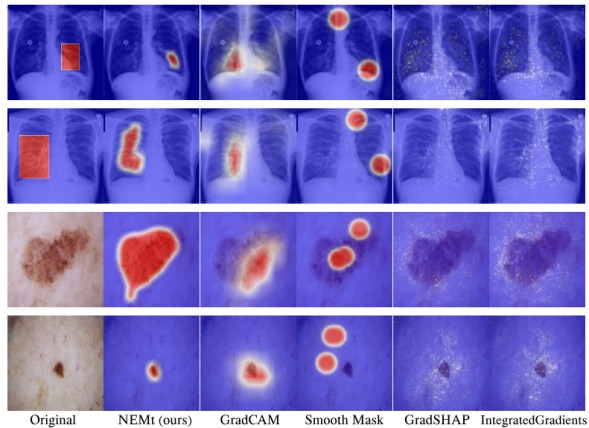

Original    NEMt (ours)    GradCAM    Smooth Mask    GradSHAP    IntegratedGradients

**Figure 3.** Different XAI methodologies explaining a RESNET50 classifier trained on the RSNA (two rows above) and HAM10000 (two rows below) datasets.

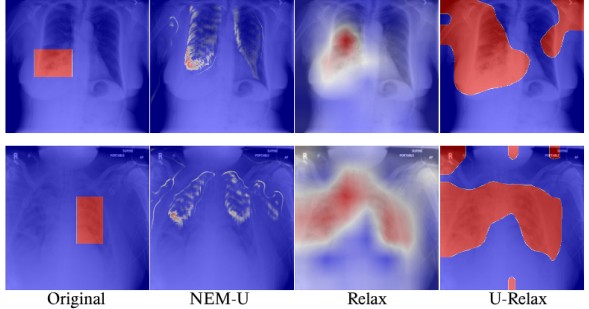

Original    NEM-U    Relax    U-Relax

**Figure 4.** Different XAI methodologies explaining a pre-trained MedCLIP model run on the RSNA dataset.

## 6 Discussion

The relevance mass results show that NEMt generated more precise masks than alternative methods (i.e., most of the predicted mask is located inside the ground truth bounding box). The superior performance in terms of complexity and sparseness indicates simple, easily understandable explanations. The 2–1000 times faster performance of NEMt makes it applicable in real-time clinical settings. In accordance with [5], RELAX is slightly better than NEM-U when looking at the relevance ranks. This can be explained by NEM's focus on a set of features and not on grading individual pixels. Considering this, it is surprising that NEMt achieved the best and second-best relevance rank results in the supervised setting. This may also explain why NEMt performed only the third best in terms of faithfulness in the skin cancer experiments, as the faithfulness score also depends on removing pixels in the right order. The strong results in both the supervised and unsupervised settings indicate that the NEM framework is a powerful general tool for XAI in the medical domain. The proposed *stochastic weighted neighborhood averaging* for artifact reduction in explanation masks made additional regularization obsolete in the supervised setting. Future work should study if it also holds in the unsupervised case. Our pro-

posed NEMt loss allows us to focus on a specific target label in contrast to the standard NEM loss (1), which would only be able to explain the **current** overall output vector of the model. This brings unique benefits, for example, it allows us to train masking units that look for a specific pathology in an input image regardless of whether the model detects it in the original input. A limitation of the current NEMt framework is the need to train a new masking network if there is a desire to explore a specific target. Future work should explore if the framework can be updated to include an input that bias the model toward a specific target during inference removing the need for retraining. A limitation of the presented study is the relatively small number of model architectures included in our experiments. Future work should study whether the results would transfer to other architectures. Furthermore, since this study has focused on medical data, it would be interesting to explore whether the NEMt framework will show the same advantages on more standard image datasets such as ImageNet [38], VOC [39], or COCO [40].

## 7 Conclusion

In this work, we have proposed **N**eural **E**xplanation **M**asks with **t**arget labels (NEMt). Our experiments indicate that it and the original NEM framework are useful for XAI in medical settings, when applied to classifiers and feature extractors, respectively. The introduction of *stochastic weighted neighborhood averaging* effectively eliminates the need for additional regularization in the supervised setting, suggesting a promising avenue for future exploration in unsupervised contexts as well. Additionally, the proposed NEMt loss provides an advantage over the original NEM loss by allowing targeting of specific labels.

## Acknowledgment

This work was supported by the Novo Nordisk Foundation under grant NFF20OC0062056 and by the Pioneer Centre for AI under DNRF grant no. P1. This work was also partially funded by the Research Council of Norway (RCN) grant no. 309439 Centre of Research-based Innovation Visual Intelligence, http://visual-intelligence.no, as well as RCN grant no. 303514.

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

# A Robustness studies

## A.1 Robustness of hyperparameter in NEMt loss function

To determine whether (2) is robust w.r.t. different choices of $\lambda_1$, we measured faithfulness [37] of NEMt models on the RSNA dataset with different choices of $\lambda_1$. The results are shown in Table A.1. Our experiments indicate that NEMtwas robust w.r.t. different hyperparameter choices.

**Table A.1.** Influence of the parameter $\lambda_1$ in the loss function (2) studied on the RSNA dataset.

| $\lambda_1$ | Faithfullness ↑ |
|---|---|
| 1 | 0.219 |
| 0.66 | 0.414 |
| 0.5 | 0.420 |
| 0.4 | 0.426 |
| 0.33 | 0.010 |

## A.2 Robustness of neighbourhood size in stochastic weighted neighborhood averaging

To determine how robust the NEMt framework is to different choices for the neighbourhood size of the stochastic weighted neighborhood averaging filter, we measured faithfulness of NEMt models trained with different neighbourhood sizes on the RSNA dataset. The experimental results are displayed in table A.2. The results indicate robustness w.r.t. this hyperparameter.

**Table A.2.** Influence of the neighbourhood size in the stochastic weighted neighborhood averaging.

| neighbourhood size | Faithfulness ↑ |
| --- | --- |
| $14 \times 14$ | 0.445 |
| $17 \times 17$ | 0.448 |
| $21 \times 21$ | 0.420 |
| $24 \times 24$ | 0.429 |
| $28 \times 28$ | 0.194 |

## A.3 Effect of using the stochastic neighbourhood averaging filter

To determine the effect of using the stochastic neighbourhood averaging filter, we have trained NEMt models on the HAM10000 data with and without using the filter and measured the faithfulness. The results are seen in table A.3. The results indicate a big improvement in faithfulness when leveraging the filter.

**Table A.3.** The effect of using the stochastic weighted neighborhood averaging filter on the HAM10000 dataset.

| | Faithfullness ↑ |
| --- | --- |
| Using filter | 0.298 |
| Not using filter | -0.001 |

