# OpenReview forum: "NEMt: Fast Targeted Explanations for Medical Image Models via Neural Explanation Masks"
_NLDL.org/2025/Conference — NLDL 2025 Oral_

### Official Review · Reviewer_kbVf · 2024-10-01
**Review of NEMt: Fast Targeted Explanations for Medical Image Models via Neural Explanation Masks**

**Confidence:** 3

**Summary:**

The work introduces NEMt, a modification of the NEM method to specifically explain image classification predictions, in contrast to the original model which explains latent representations.
The work also extends the NEM method with a new final layer to remove artifacts dubbed "stochastic weighted neighborhood averaging" which was needed as standard NEM produced a lot of artifacts in classification explanation.
NEMt works by essentially training a segmentation network that explains a given classification network by outputting an attribution score for each pixel.

During inference, the image is predicted by the classification network and the segmentation network is given the image and some representations from the classification network and predicts the attribution map.
During training additional steps are applied where the pixels indicated by the attribution map are occluded and the new image is fed through the classification network.
The segmentation network is then trained to find the attributions that maximize the difference in prediction while a regularization loss forces it to segment a smaller area.

The method is compared to several others using many relevant metrics in the new classification setting as well as the original latent space setting.
The results show that NEMt is still competitive in the original latent space setting, though no comparison to the original NEM is made which leaves it uncertain whether the new artifact removal is useful in this case.
The results also show that NEMt is better than the other methods in the classification setting for most metrics on two different datasets.

**Strengths:**

The work is well written and contains all immediately relevant details.
The NEM method is shown to be very useful in the classification setting and promises to be a weakly supervised segmentation method as well, though this is not mentioned in the work.
The artifact removal procedure is shown, though not thoroughly, to be helpful.
Additionally, artifacts are a general issue with attribution maps which I am glad to see tackled.
The evaluation is thorough with many metrics and previous methods, though missing some relevant ones.

In short, the work improves an already promising method which it presents clearly and evaluates thoroughly.

**Weaknesses:**

The NEM and NEMt methods do share a lot of similarities with methods developed by the Weakly Supervises Semantic Segmentation (WSSS) field which has also developed methods to extract masks using pretrained classification models, however, these are not mentioned at all in work.
The artifact removal is described as useful and necessary but its impact is not shown through evaluation, something that could have been achieved by evaluating how the method performs without it or with the original NEM regularizing loss instead (especially since one of the test cases is specifically in the domain NEM was developed for).
NEMt is compared against RELAX in the latent space setting. Since RELAX can be described to be a latent space adaption of RISE, it is strange to not also compare it against RISE in the classification setting.
While relevance mass is a useful metric it is biased towards metrics that produce positive attributions for relevant areas and negative otherwise, this is not strictly speaking needed for an attribution. This can be alleviated with normalization, but whether this is done is not described.

**Final Rebuttal Confidence:**

4

**Final Rebuttal Justification:**

The original work contributes an evaluation of a novel method (NEM) on a new problem (image classification). Additionally, the method introduced novel solutions (artifact removal) needed for the method to work in the new domain and might be generally useful for other similar methods. There were some missing details like comparisons to similar ideas in other domains, ablation evaluation of their new solutions, a relevant existing method not being compared to, and a tiny bit of implementation details missing. These were all clarified in the rebuttal and promised to be included in the revised version. For these reasons, the work is a clear fit for the conference.

**Justification:**

While there are some weaknesses with the study, they mostly have to do with how the method is presented and evaluated, not with the method itself which is sound and well-evaluated.
The work is easy to understand, its contribution is clear, and may even be relevant beyond the XAI domain as a WSSS method.
The work is therefore not only interesting to XAI researchers but computer vision and ML in general which is a great fit for the conference.

---

> ### Author Rebuttal · Authors · 2024-10-22
>
> # Weaknesses
> ## 1. WSSS mentions
> In this work, we have focused on the realm of XAI and since WSSS is not strictly about generating model explanation, We choose not to focus on this. We can now see that there are overlaps between the two fields and thus we have updated the text to reflect this.
> ## 2. Artifact removal system
> The reason, we have not delved deeper into the original B term for supervised classification is, that our experiments indicate that it breaks down in the supervised case. Why this happens, is not exatcly clear to us, which is why we did not delve into this further in the paper. There are two ways we tried to implement the negative distance of the B term and both appear to fail.
> The first is to move the negated masked vector away from the masked vector but this does not really work as intented. This is potentially due to the compression in the output layer ( due to the softmax)  The other would be to move the target logit of the negated masked vector away from the masked vector, but for some reason our models would not converge with this option.
>
> We will revise the text to better reflect this issue.
>
> To show the effect with and without the smoothing filter, we have calculated the faithfullness metrics for the NEMT with and without smoothing filter on the lesion dataset:
>
> To show the effect with and without the smoothing filter, we have calculated the faithfullness metrics for the NEMT with and without smoothing filter on the lesion dataset:
>
> |                                     | Faithfullness |
> |-------------------------------------|---------------|
> | With stochastic smoothing filter    | -0.001        |
> | Without stochastic smoothing filter | 0.298         |
>
> We will include this in the final paper in the appendix
>
> ## 3. RISE
> Good idea.
> We have run the experiments for rise and will add them to our manuscript.
> The Rise method has the second third highest faithfullness on the Pneumonia dataset,
> and the second lowest faithfullness on the skin lesion dataset. Please see manuscript for overall comparison.
>
> For the Pneumonia dataset we get:
> | Method | Relevancy Rank | Relevancy Mass | Complexity | Sparseness | Faithfullness | Time  |
> |--------|----------------|----------------|------------|------------|---------------|-------|
> | NEMt   | 0.215          | 0.192          | 9.343      | 0.744      | 0.419         | 0.003 |
> | RISE   | 0.124          | 0.077          | 10.686     | 0.274      | 0.24          | 7.368 |
>
> For the lesion dataset we get:
> | Method | Relevancy Rank | Relevancy Mass | Complexity | Sparseness | Faithfullness | Time  |
> |--------|----------------|----------------|------------|------------|---------------|-------|
> | NEMt   | 0.723          | 0.838          | 8.855      |  0.828      |  0.298       | 0.003 |
> | RISE   | 0.271          | 0.279          | 10.675     |  0.290      | -0.009       | 7.368 |
>
> ## 4. Normalization of relevancy mass
> All attributions were minmax normalized before generating experimental results. We will update the manuscript to state this.

---

### Official Review · Reviewer_iqmS · 2024-10-06
**An interesting XAI module for artificial neural networks**

**Confidence:** 3

**Summary:**

The paper proposes a variant to a recently introduced Neural Explanation Mask (NEM) framework, called NEMt, where the additional "t" in the name corresponds to the notion of "target labels".  The idea is that the original NEM framework works in an unsupervised setting, whereas the proposed framework works for artificial neural networks that operate in a supervised manner. The authors experimentally validate the performance of their approach using two datasets from the medical domain and moreover evaluate the performance of the proposed method using several metrics (relevance rank, relevance mass, complexity, sparseness) that have been introduced and provide justification for the "goodness" of the provided explanations.  Furthermore, the method is very lightweight as exhibited by the experiments, thus providing explanations in time comparable to one baseline method (Grad Shape) or, faster compared to the other baselines explored (Smooth Pixel Mask, GradCAM, Integrated Gradients).

**Strengths:**

- A new method for XAI thus has good properties on several metrics (including the time to provide an explanation).

- The proposed method works on top of existing trained models, thus making it very appealing for adoption in various settings.

**Weaknesses:**

- No source code provided.  I would urge the authors to provide a link with the source code for their experiments upon acceptance of their work.  This can only benefit the community and the authors.

- Other than the above, I cannot think of any real weaknesses for this work from a scientific point of view.  Even though the novelty is not very high since it relies on the NEM framework, nevertheless, the idea is natural and very much important since supervised neural networks are used in so many contexts.

Finally, below are some comments for the presentation.

- Figure 1 seems to be out of order and I do not believe it is referenced from the main text.

- In Section 3.1 where you describe the NEM framework I was a little bit confused in retrospect.  In line 137 you say that the output of model $\Phi$ is $y$, but what is really $y$?  If we are talking about unsupervised learning, then this $y$ cannot be a label - and $y$ is (almost?) universally accepted as a "label"; so, most likely you want to output the identify function (e.g., as in an auto encoder) or some sort of embedding of the original input?  Please note that I was not familiar with the NEM framework before reading this paper and perhaps this is why it is not so clear to me what you are trying to say and how that would fit in an "unsupervised" setting where NEM operates.  Along these lines, in line 170, you mention that $d$ is some distance function.  Given that $d$ takes as input such $y$'s, it would be nice if you can provide an example of the kind of functions that you have in mind.

The above weaknesses are the reason that I am giving a rating of 4 for this work and not a 5.

**Final Rebuttal Confidence:**

4

**Final Rebuttal Justification:**

Reading the reviews of others has increased my confidence on the rating of the paper.  I believe that this is a paper that has sufficient novelty and should be accepted, especially given the changes and small additions that the reviewers have requested and the authors have promised to include in the final version of the paper.

**Justification:**

The paper deals with an important problem; that of providing explanations in real-world applications of machine learning where artificial neural networks are used.  Furthermore, an important aspect of the approach is the modularity of the proposed method, since it can be applied on any pre-trained artificial neural network (and such applications are pretty everywhere nowadays).  Overall, I think the paper is well-written and good discussion has been made throughout.  Important baselines are being used and a nice set of metrics is also used for categorizing the method, with respect to other methods.

---

> ### Author Rebuttal · Authors · 2024-10-22
>
> # Weaknesses
> ## 1. Missing Source code
> Very good point.
> It is true that the community should strive to publish open source. But we do not wish to break confidentiality. This is why we have created a anonymized github repo and linked to it in the abstract of the report.
> The final line in our paper contains a URL, but maybe this was not obvious enough. We have now altered it to make it more obvious. Please see the final part of the abstract in the revised manuscript.
> ## 2. Figure 1 out ouf order
> Thank you for pointing this out. We had included a reference to the figure on line 144 of the manuscript, but it is clear now that there is a disconnect between figure placement and model reference. Thus we have moved the figure to be closer  and after the reference for clarity.
> ## 3. Use of y
> Thank you for the comment.
> We choose to use y for the output of the unsupervised network for consistency, as this is what was done in the original NEM-U paper.

---

### Official Review · Reviewer_z9b3 · 2024-10-08
**Organized paper that expands on existing XAI framework and shows results on two common medical imaging datasets**

**Confidence:** 3

**Summary:**

The paper proposes Neural Explanation Masks with target labels (NEMt), an XAI method which is a variant of the Neural Explanation Mask (NEM) framework. The NEM framework trains an explanation module for self-supervised representations that outputs explanation masks for the parts of the input that influence the representation most. In this article, the authors have adapted the method to

The NEMt loss function differs from NEM by optimizing for the mask that yields the largest change in the predicted class logit, rather than optimizing for the mask that yields the most similar representation. Otherwise, both methods encourage sparse masks by $\ell_1$ regularization. NEMt enforces binary masks by removing artifacts with a stochastic smoothing filter via weighted neighborhood averaging. This is done during the training of the masking network. Originally, the NEM framework encourages binary masks by adding a penalty term to the loss function, but this term is absent and instead replaced by the stochastic smoothing filter in NEMt.

The method is evaluated on two datasets - the 2018 RSNA Pneumonia Detection 250 Challenge dataset and the HAM10000 dataset -using established quantitative metrics for XAI.

**Strengths:**

- The objective is clear and based on established work
- The method is evaluated using established XAI metrics for explanation quality
- The work is novel in the sense that it expands an established framework to apply to the supervised setting with labels
- The paper is organized, and the text is well written
- The experiments show improvement against existing XAI methods on two medical imaging datasets.

**Weaknesses:**

- Although this work focuses on the application of the method to medical data, the results would have been more significant if NEMt was evaluated on other image datasets as well. In particular, VOC and COCO to compare results with previous work. This could also allow comparison of the performance of the method across various models, e.g. vision transformers. Currently, the results are only shown for frozen ResNet50 architectures.
- Lacking ablations on choice of $\lambda_1$ and $\lambda_2$ and how stable the method is to changes of the training loss parameters
- I wonder if the problem with artifacts which appeared in the initial experiments without the stochastic smoothing filter was due to the choice of the $B(x)$ regularization term, and if the authors have tried different choices?
- The authors do not mention the neighborhood size used for the smoothing filter or if this was ablated on

**Justification:**

The paper proposes an extension of an established framework, which shows promising results in the medical setting. I hope the authors can address the above weaknesses to justify the completeness and thoroughness of their work.

---

> ### Author Rebuttal · Authors · 2024-10-22
>
> ## 1. Weakness (Dataset and model ablation)
> It is a fair point, that we should have included more dataset such as VOC and COCO and additional models such as vision transformers. Unfortunately, given the timeframe for the rebutal phase, we do not have time to do this in a manner, where we can garantee the quality of the experimental results. We have included a discussion of these limitations in the discussion section of our paper.
>
>
> ## 2. Weakness (Loss function ablation)
> For ablation of the NEM loss function, we refer to the original NEM-U paper, which indicates robustness to loss term perturbation.
> For our proposed NEMt loss, we have now included a ablation study of the hyperparameter lambda_1 on the pneumonia dataset. This will be included in the final paper and can be seen below.
> |                   lambda_1                  | Faithfullness |
> |-------------------------------------|---------------|
> | 1                                         |  0.219       |
> | 1/1.5                                   |  0.414       |
> |  1/2                                     | 0.420        |
> | 1/2.5                                   |  0.426       |
> | 1/3                                      | 0.0104      |
>
>
>
> ## 3. Weakness (Exploration of B(x))
>
> The reason, we have not delved deeper into the original B term for supervised classification is, that our experiments indicate that it breaks down in the supervised case. Why this happens, is not exatcly clear to us, which is why we did not delve into this further in the paper. There are two ways we tried to implement the negative distance of the B term and both appear to fail.
> The first is to move the negated masked vector away from the masked vector but this does not really work as intented. This is potentially due to the compression in the output layer ( due to the softmax)  The other would be to move the target logit of the negated masked vector away from the masked vector, but for some reason our models would not converge with this option.
>
> We will revise the text to better reflect this issue.
>
> To show the effect with and without the smoothing filter, we have calculated the faithfullness metrics for the NEMT with and without smoothing filter on the lesion dataset:
>
> |                                                        | Faithfullness |
> |-----------------------------------------------|------------------|
> | Without stochastic smoothing filter | -0.001          |
> | With stochastic smoothing filter      | 0.298           |
>
> We will include this in the final paper in the appendix
>
> ## 4. The neighbourhood size of the smoothing filter
> Our smoothing filter used a size 21 x 21 kernel. We have now created an abblation study of various kernel size. This is run on the pneumonia dataset.  This can be seen below and will be included in the final paper.
> |              kernel size              | Faithfullness |
> |-------------------------------------|-------------------|
> | 14 x 14                               | 0.445             |
> | 17 x 17                               | 0.448             |
> | 21 x 21                               | 0.420             |
> | 24 x 24                               | 0.429             |
> | 28 x 28                               | 0.194             |

---

### Meta-Review · Area_Chair_eGap · 2024-11-03

**Recommendation:** Accept (Oral)
**Confidence:** 4

**Metareview:**

The paper proposes a method called Neural Explanation Masks with target labels (NEMt), an extension of the existing NEM framework tailored for supervised settings to improve interpretability for neural nets. The method addresses a weakness of the existing NEM and optimizes for target labels, incorporates a stochastic smoothing filter to remove artifacts to provide more reliable explanations for classification. The paper demonstrates improvements over existing baselines across multiple metrics. The authors tested the approach on two datasets and provided a code repository for reproducibility. All reviewers regarded the paper as an interesting contribution but also raised some comments. The reviewers noted that the paper would benefit from a broader evaluation on additional datasets, comparison with more related methods and more details on certain implementation aspects. Most of the comments from the reviewers were addressed during the rebuttal and the authors added some more experiments and ablation studies and gave some additional details for clarification which they plan to add to the manuscript prior to publication. This was well received by the reviewers. Given all strengths and weaknesses, all reviewers rated the paper very positively and suggested accepting it for NLDL.

**Suggested Changes To The Recommendation:**

1: I agree that the recommendation could be moved down

---

### Decision · Program_Chairs · 2024-11-06

**Decision:**

Accept (Oral)

**Comment:**

We recommend an oral and a poster presentation given the AC and reviewers recommendations.